# SEMI-IMPLICIT BACK PROPAGATION

## ABSTRACT

Neural network has attracted great attention for a long time and many researchers are devoted to improve the effectiveness of neural network training algorithms. Though stochastic gradient descent (SGD) and other explicit gradient-based methods are widely adopted, there are still many challenges such as gradient vanishing and small step sizes, which leads to slow convergence and instability of SGD algorithms. Motivated by error back propagation (BP) and proximal methods, we propose a semi-implicit back propagation method for neural network training. Similar to BP, the difference on the neurons are propagated in a backward fashion and the parameters are updated with proximal mapping. The implicit update for both hidden neurons and parameters allows to choose large step size in the training algorithm. Finally, we also show that any fixed point of convergent sequences produced by this algorithm is a stationary point of the objective loss function. The experiments on both MNIST and CIFAR-10 demonstrate that the proposed semi-implicit BP algorithm leads to better performance in terms of both loss decreasing and training/validation accuracy, compared to SGD and a similar algorithm ProxBP.

## 1 INTRODUCTION

Along with the rapid development of computer hardware, neural network methods have achieved enormous success in divers application fields, such as computer vision (Krizhevsky et al., 2012), speech recognition (Hinton et al., 2012; Sainath et al., 2013), nature language process (Collobert et al., 2011) and so on. The key ingredient of neuron network methods amounts to solve a highly non-convex optimization problem. The most basic and popular algorithm is stochastic gradient descent (SGD) (Robbins & Monro, 1951), especially in the form of "error" back propagation (BP) (Rumelhart et al., 1986) that leads to high efficiency for training deep neural networks. Since then many variants of gradient based methods have been proposed, such as Adagrad(Duchi et al., 2011), Nesterov momentum (Sutskever et al., 2013), Adam (Kingma & Ba, 2014) and AMSGrad (Reddi et al., 2019). Recently extensive research are also dedicated to develop second-order algorithms, for example Newton method (Orr & Müller, 2003) and L-BFGS (Le et al., 2011).

It is well known that the convergence of explicit gradient descent type approaches require sufficiently small step size. For example, for a loss function with Lipschitz continuous gradient, the stepsize should be in the range of $(0, 2/\mathcal{L})$ for $\mathcal{L}$ being the Lipschitz constant, which is in general extremely big for real datasets. Another difficulties in gradient descent approaches is to propagate the "error" deeply due to nonlinear activation functions, which is commonly known as gradient vanishing. To overcome these problems, implicit updates are more attractive. In (Frerix et al., 2018), proximal back propagation, namely ProxBP, was proposed to utilize the proximal method for the weight updating. Alternative approach is to reformulate the training problem as a sequence of constrained optimization by introducing the constraints on weights and hidden neurons at each layer. Block coordinate descent methods (Carreira-Perpinan & Wang, 2014; Zhang & Brand, 2017) were proposed and analyzed to solve this constrained formulation with square loss functions. Along this line, the Alternating direction method of multipliers (ADMM) (Taylor et al., 2016; Zhang et al., 2016) were also proposed with extra dual variables updating.

Motivated by proposing implicit weight updates to overcome small step sizes and vanishing gradient problems in SGD, we propose a semi-implicit scheme, which has similar form as "error" back propagation through neurons, while the parameters are updated through optimization at each layer. It can be shown that any fixed point of the sequence generated by the scheme is a stationary point of the

objective loss function. In contrast to explicit gradient descent methods, the proposed method allows to choose large step sizes and leads to a better training performance per epoch. The performance is also stable with respect to the choice of stepsizes. Compared to the implicit method ProxBP, the proposed scheme only updates the neurons after the activation and the error is updated in a more implicit way, for which better training and validation performances are achieved in the experiments on both MNIST and CIFAR-10.

## 2 NOTATIONS

Given input-output data pairs $(X, Y)$, we consider a $N$-layer feed-forward fullly connected neural network as shown in Figure 1. Here, the parameters from the $i$-th layer to the $(i+1)$-th layer are the

$$X \xrightarrow{W_1, b_1} G_2 \xrightarrow{\sigma} F_2 \xrightarrow{W_2, b_2} G_3 \cdots G_{N-1} \xrightarrow{\sigma} F_N\text{-}Y^{W_{N-1}, b_{N-1}} \xrightarrow{} F_N$$

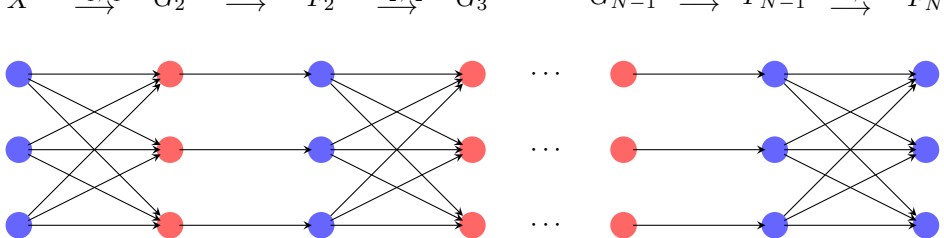

Figure 1: N-layer Neural Network

weight matrix $W_i$ and bias $b_i$, and $\sigma$ is a non-linear activation function, such as sigmod or ReLU. We denote the neuron vector at $i$-th layer before activation as $G_i$, and the neurons after activation as $F_i$, i.e. $F_1 = X$,

$$G_{i+1} = W_i F_i + b_i;$$
$$F_{i+1} = \sigma(G_{i+1}) \tag{1}$$

for $i = 1, \cdots, N-1$. We note that in general at the last layer, there is no non-linear activation function and $F_N = G_N$. For ease of notation, we can use an activation function $\sigma$ as identity. The generic training model aims to solve the following minimization problem:

$$\min_\theta J(\theta; X, Y) := L(F_N, Y) \tag{2}$$

where $\theta$ denotes the collective parameter set $\{W_i, b_i\}_{i=1}^{N-1}$ and $L$ is some loss function.

## 3 SEMI-IMPLICIT BACK-PROPAGATION METHOD

In the following, we first present the classic back propagation (BP) algorithm for an easier introduction of the proposed semi-implicit method.

### 3.1 BACK PROBATION METHOD

The widely used BP method Rumelhart et al. (1986) is based on gradient descent algorithm:

$$\theta^{k+1} = \theta^k - \eta \frac{\partial J(\theta^k; X, Y)}{\partial \theta} \tag{3}$$

where $\eta > 0$ is the stepsize. The main idea of BP algorithm is to use an efficient error propagation scheme on the hidden neurons for computing the partial derivatives of the network parameters at each layer. The so-called "error" signal $\delta_N \equiv \frac{\partial J}{\partial F_N}$ at the last level is propagated to the hidden neurons $\delta_i \equiv \frac{\partial J}{\partial F_i}$ using the chain rule. In fact for a square loss function, the gradient at the last layer $\delta_N = \frac{\partial L(F_N, y)}{\partial F_N} = F_N - y$ is indeed an error. The propagation from $\delta_{i+1}$ to $\delta_i$ for $i = N-1, \cdots, 1$ is then calculated as

$$\delta_i : \frac{\partial J}{\partial F_i} = \frac{\partial G_{i+1}}{\partial F_i} \frac{\partial F_{i+1}}{\partial G_{i+1}} \frac{\partial J}{\partial F_{i+1}} = W_i^T \left( \frac{\partial \sigma(G_{i+1})}{\partial G_{i+1}} \odot \delta_{i+1} \right). \tag{4}$$

And the partial derivative to $W_i$ can be computed as

$$\frac{\partial J}{\partial W_i} := \frac{\partial G_{i+1}}{\partial W_i}\frac{\partial F_{i+1}}{\partial G_{i+1}}\frac{\partial J}{\partial F_{i+1}} = (\frac{\partial \sigma(G_{i+1})}{\partial G_{i+1}} \odot \delta_{i+1})F_i^T. \tag{5}$$

At $k-$th iteration, after a forward update of the neurons $F_i^k$ by (1) using the current parameters sets $\{W_i^k, b_i^k\}, i = 1, \cdots, N-1$, we can compute the "error" signal at each neurons sequentially from the $i+1$ level to $i$ level by (4) and the parameters $W_i^{k+1}$ is updated according to the gradient at the point $W_i^k$ computed by (5).

## 3.2 SEMI-IMPLICIT UPDATES

Compared to the BP method, we propose to update the hidden neurons and the parameters sets at each layer in an implicit way.

At the iteration $k$, given the current estimate $\theta^k : \{W_i^k, b_i^k\}$, we first update the neuron $F_i^k$ and $G_i^k$ in a feedforward fashion as BP method, by using (1) for $i = 1, \cdots, N-1$. For the backward stage, we start with updating neuron $F_N$ at the the last layer using the gradient descent:

$$\delta_N^k = \frac{\partial L(F_N^k, Y)}{\partial F_N}, \quad F_N^{k+\frac{1}{2}} = F_N^k - \eta\delta_N^k. \tag{6}$$

For $i = N-1, \cdots, 1$, given $F_{i+1}^{k+\frac{1}{2}}$, the parameters $W_i, b_i$ are updated by solving the following optimization problem (once)

$$\begin{cases} W_i^{k+1} = \underset{W_i}{\arg\min}\|\sigma(W_iF_i^k + b_i^k) - F_{i+1}^{k+\frac{1}{2}}\|_F^2 + \frac{\lambda}{2}\|W_i - W_i^k\|_F^2 \\ b_i^{k+1} = \underset{b_i}{\arg\min}\|\sigma(W_i^{k+1}F_i^k + b_i) - F_{i+1}^{k+\frac{1}{2}}\|_F^2 + \frac{\lambda}{2}\|b_i - b_i^k\|_F^2 \end{cases} \tag{7}$$

where $\lambda > 0$ is a parameter that is corresponding to stepsize. This update of parameters is related to using an implicit gradient based on so-called proximal mapping. Taking $W_i$ as an example, the optimality in (7) gives

$$W_i^{k+1} = W_i^k - \frac{1}{\lambda}\nabla f(W_i^{k+1})$$

where $f(W_i) = \|\sigma(W_iF_i^k + b_i^k) - F_{i+1}^{k+\frac{1}{2}}\|_F^2$. Compared to a direct gradient descent step, this update is unconditionally stable for any stepsize $1/\lambda$. We note that proximal mapping was previously proposed for training neural network as ProxBP in (Frerix et al., 2018). However the update of the parameter sets is different as ProxBP uses $G_{i+1}$ for the data fitting at each layer. The two subproblems at each layer can be solved by a nonlinear conjugate gradient method.

After the update of $W_i^{k+1}$ and $b_i^{k+1}$, we need to update the hidden neuron $F_i$. As classical BP, we first consider the gradient at $F_i^k$ as

$$\frac{\partial J}{\partial F_i^k} = \frac{\partial F_{i+1}^k}{\partial F_i^k}\frac{\partial J}{\partial F_{i+1}^k} = \frac{\partial G_{i+1}^k}{\partial F_i^k}\frac{\partial F_{i+1}^k}{\partial G_{i+1}^k}\frac{\partial J}{\partial F_{i+1}^k}. \tag{8}$$

It can be seen that the partial derivative $\frac{\partial G_{i+1}^k}{\partial F_i^k} := W_i^k$. Different from BP and ProxBP, we use the newly updated $W_i^{k+1}$ instead of $W_i^k$ to compute the error:

$$\delta_i^k := (W_i^{k+1})^T \cdot (\partial\sigma(G_{i+1}^k) \odot \delta_{i+1}^k), \quad F_i^{k+\frac{1}{2}} = F_i^k - \eta\delta_i^k \tag{9}$$

By this formula, the difference can be propagated from the level $N$ to 1. At the last level $i = 1$, we only need to update $W_1^{k+1}$ and $b_1^{k+1}$ as $F_1 = X$. The overall semi-implicit back propagation method is summarized in Algorithm 1. For large scale training problem, the back propagation is used in the form of stochastic gradient descent (SGD) using a small set of samples. The proposed semi-implicit method can be easily extended to stochastic version by replacing $(X, Y)$ by a batch set $(X_{\text{mini}}, Y_{\text{mini}})$ at each iteration in Algorithm 1.

---

**Algorithm 1** Semi-implicit back propagation

---

**Input:** Current parameters $\theta^k = \{W_i^k, b_i^k\}$
*// Forward pass*
$F_1 = X$
**for** $i = 1$ **to** $N - 1$ **do**
  $G_{i+1}^k = W_i^k F_i^k + b_i$
  $F_{i+1}^k = \sigma(G_{i+1}^k)$
**end for**
*// Update on the hidden neurons and the parameters*
Update on $F_N^k$
**for** $i = N - 1$ **to** $2$ **do**
  Implicit update on $W_i^k, b_i^k$
  Error propagation and update on $F_i^k \to F_i^{k+\frac{1}{2}}$
**end for**
Implicit update on $W_1^k, b_1^k$
**Output:** New parameters $\theta^{k+1} = \{W_i^{k+1}, b_i^{k+1}\}$

---

## 3.3 FIXED POINTS OF SEMI-IMPLICIT METHOD

The follow proposition indicates that any fixed point of the iteration is a stationary point of the objective energy function.

**Proposition 1** *Assume that $L$ and the activation functions $\sigma$ are continuously differentiable. If $\theta^k \xrightarrow{k \to \infty} \theta^*$ for $\theta^* = \{W_i^*, b_i^*\}$, then $\theta^*$ is a stationary point of the energy function $J(\theta; X, Y)$.*

*Proof* Due to the forward update, $\{W_i^k, b_i^k\} \xrightarrow{k \to \infty} \{W_i^*, b_i^*\}$ infers that $\{F_i^k, G_i^k\} \xrightarrow{k \to \infty} \{F_i^*, G_i^*\}$ where $G_i^* = W_i^* F_{i-1}^* + b_i^*$ and $G_i^* = \sigma(F_i^*)$ for $i = 1, \cdots, N$. At the last layer, the neuron $F_N$ is updated with gradient descent:

$$F_N^{k+\frac{1}{2}} = F_N^k - \eta \frac{\partial L(F_N, Y)}{\partial F_N^k} \tag{10}$$

Take a limit, we have

$$\lim_{k \to \infty} F_N^{k+\frac{1}{2}} = \lim_{k \to \infty} F_N^k - \eta \frac{\partial L(F_N, Y)}{\partial F_N^k} \tag{11}$$

$$= F_N^* - \eta \frac{\partial J}{\partial F_N^*} \tag{12}$$

Now we show

$$\lim_{k \to \infty} F_i^{k+\frac{1}{2}} = F_i^* - \eta \frac{\partial J}{\partial F_i^*}; \quad \frac{\partial J}{\partial W_i^*} = 0 \tag{13}$$

for $i = N, \cdots, 2$ using mathematical induction. The first equation is shown for $i = N$. By the optimality of equation 7, we have

$$\frac{\lambda}{2}(W_i^{k+1} - W_i^k) = [\partial \sigma(W_i^{k+1} F_i^k + b_i^k) \odot (F_{i+1}^{k+\frac{1}{2}} - \sigma(W_i^{k+1} F_i^k + b_i^k))](F_i^k)^T \tag{14}$$

Let $k \to \infty$, we obtain

$$
\begin{aligned}
0 &= [\partial\sigma(W_i^* F_i^* + b_i^*) \odot (F_{i+1}^* - \eta\frac{\partial J}{\partial F_{i+1}^*} - \sigma(W_i^* F_i^* + b_i^*))](F_i^*)^T \\
&= [\partial\sigma(G_{i+1}^*) \odot (-\eta\frac{\partial J}{\partial F_{i+1}^*})](F_i^*)^T \\
&= -\eta[\partial\sigma(G_{i+1}^*) \odot \frac{\partial J}{\partial F_{i+1}^*}](F_i^*)^T \\
&= -\eta\frac{\partial G_{i+1}^*}{\partial W_i^*}\frac{\partial F_{i+1}^*}{\partial G_{i+1}^*}\frac{\partial J}{\partial F_{i+1}^*} \\
&= -\eta\frac{\partial J}{\partial W_i^*}
\end{aligned}
\tag{15}
$$

It is easy to see that the limit of $F_i^{k+\frac{1}{2}}$ for $i = N-1, \cdots, 1$ is:

$$
\begin{aligned}
\lim_{k\to\infty} F_i^{k+\frac{1}{2}} &= \lim_{k\to\infty} F_i^k - (W_i^{k+1})^T[\partial\sigma(G_{i+1}^k) \odot (F_{i+1}^k - F_{i+1}^{k+\frac{1}{2}})] \\
&= F_i^* - \eta(W_i^*)^T[\partial\sigma(G_{i+1}^*) \odot \frac{\partial J}{\partial F_{i+1}^*}] \\
&= F_i^* - \eta\frac{\partial G_{i+1}^*}{\partial F_i^*}\frac{\partial F_{i+1}^*}{\partial G_{i+1}^*}\frac{\partial J}{\partial F_{i+1}^*} \\
&= F_i^* - \eta\frac{\partial J}{\partial F_i^*}
\end{aligned}
\tag{16}
$$

With mathematical induction we obtain that $\frac{\partial J}{\partial \theta^*} = \mathbf{0}$. $\qquad\square$

## 4    NUMERICAL EXPERIMENTS

In this section, we will compare the performance of BP, ProxBP Frerix et al. (2018) and the proposed semi-implicit BP using MNIST and CIFAT-10 datasets. All the experiments are performed on MATLAB with a 12-core CPU and the same network setting and initializations are used for a fair comparison. We use softmax cross-entropy for the loss function $L$ and ReLU for the activation function, as usually chosen in classification problems. For the linear CG used in ProxBP and nonlinear CG in semi-implicit BP, the iterations number is set as 5. Finally the weights and bias are initialized by normal distribution with average 0 and standard deviation 0.01.

In MNIST experiments, the training set contains 55000 samples and the rest are included in the validation set. For CIFAR-10, a set of 45000 samples is used as training set and the rest as validation set. The training process are performed 5 times, and Table 1 shows the average training and validation accuracy achieved by SGD, Semi-implicit BP with different learning rates $\eta$ (for SGD) and $1/\lambda$ for semi-implicit BP on MNIST. It can be seen that after 2 epoch, semi-implicit BP method already achieves high accuracy as high as 99% and the performance is very stable with respect to different stepsize choices, while SGD fails for some choices of stepsize $\eta$. For ProxBP, we present the results with $\eta = 1$ as the best performance is achieved with this set of parameters. The highest accuracies are marked in bold in each column, and we can see that semi-implicit BP achieves the highest training and test accuracy than BP and ProxBP.

In Figure 2 and 3, we show the performance of the three methods per epoch for MNIST and CIFAR-10. For the CIFAR-10 experiment, we train a $3072 \times 2000 \times 500 \times 10$ neural network and the rest settings are the same as MNIST dataset experiments. For both datasets, we choose the step size $\eta = 0.1$ for semi-implicit BP and ProxBP, while a smaller one to guarantee SGD achieves a best better performance. The evolution of training loss and training accuracy shows that the performance of ProxBP is not as good as SGD for MNIST while it is better than SGD on CIFAR-10. For both experiments, the performance of semi-implicit BP method leads to the fastest convergence. The improvement on the validation accuracy also demonstrates that the proposed semi-implicit BP method also generalize well in a comparison to the other two methods.

| MNIST($784 \times 500 \times 10$) | Training/validation accuracy | | | | |
|---|---|---|---|---|---|
| learning rates | 100 | 10 | 1 | 0.1 | 0.01 |
| SGD, $\eta$ | 0.0985 | 0.1123 | 0.0980 | 0.9487 | 0.8885 |
| | 0.1002 | 0.1126 | 0.0924 | 0.9492 | 0.8988 |
| ProxBP ($\lambda = 1, \eta$) | 0.9239 | 0.9349 | 0.9390 | 0.9032 | 0.8415 |
| | 0.9344 | 0.9374 | 0.9494 | 0.9244 | 0.8832 |
| ProxBP ($\eta = 1, 1/\lambda$) | 0.9420 | 0.9444 | 0.9383 | 0.9171 | 0.8743 |
| | 0.9486 | 0.9554 | 0.9494 | 0.9344 | 0.9064 |
| Semi-implicit ($\lambda = 1, \eta$) | **0.9780** | **0.9801** | **0.9904** | **0.9737** | 0.9104 |
| | 0.9710 | 0.9724 | **0.9800** | **0.9748** | 0.9338 |
| Semi-implicit | 0.9765 | 0.9752 | 0.9735 | 0.9598 | **0.9206** |
| ($\eta = 0.1, 1/\lambda$) | **0.9736** | **0.9778** | 0.9738 | 0.9672 | **0.9394** |

Table 1: Training and validation accuracy with different step sizes for 2 epoch.

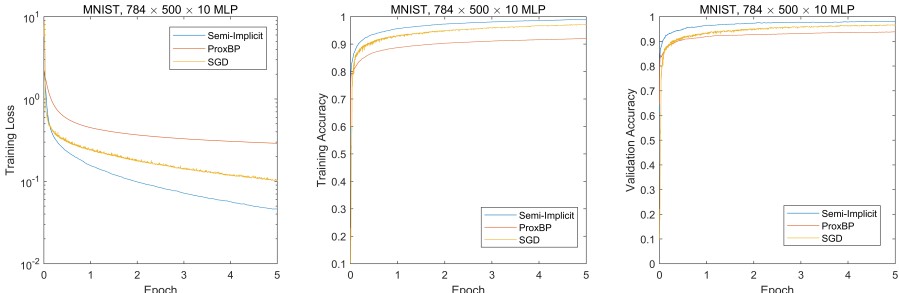

Figure 2: MNIST. Batch size 100

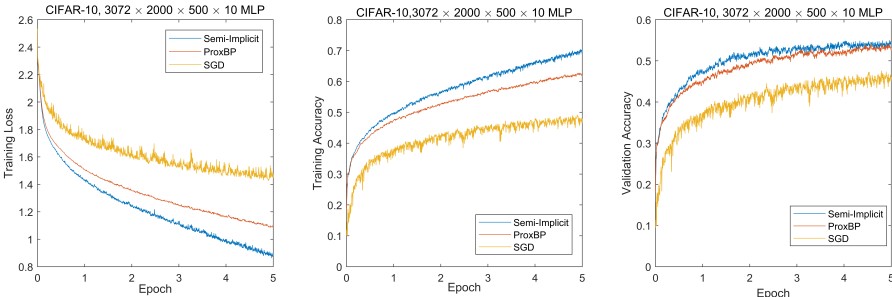

Figure 3: CIFAR-10. Batch size 100

## 5 CONCLUSION

We proposed a novel optimization scheme in order to overcome the difficulties of small stepsize and vanishing gradient in training neural networks. The computation of new scheme is in the spirit of error back propagation, with an implicit updates on the parameters sets and semi-implicit updates on the hidden neurons. The experiments on both MNIST and CIFAR-10 show that the proposed semi-implicit back propagation has better performance per epoch compared to SGD and ProxBP. It is demonstrated in the experiment that larger step sizes can be adopted without losing stability and performance. It can be also seend that the proposed scheme is flexible and some regularization can be easily integrated if needed. The fixed points of the scheme are shown to be stationary points of the objective loss function and further rigorous theoretical convergence will be explored in an ongoing work.

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
