# OpenReview forum: "Semi-Implicit Back Propagation"
_ICLR.cc/2020/Conference — Reject_

### Official Review · AnonReviewer1 · 2019-10-21
**Official Blind Review #1**

**Rating:** 1

**Review:**

The work is based on the recent paper 'Proximal Backpropagation', Frerix et al., ICLR 2018, which views error back propagation as block coordinate gradient descent steps on a penalty functional comprised of activations and parameters.

Instead of taking proximal steps on the linear layers as in (Frerix et al., 2018), the authors also pull the non-linearity \sigma into the proximal steps. Another interesting deviation is the idea to consider the newly updated weights {W_i}^{k+1} when updating the activations F_i^{k+1} in the backward pass.

While potentially offering a faster convergence with respect to epochs, the nonlinear updates have two major drawbacks:

1) While there are preliminary theoretical results (fixed points of the method are critical points), it remains unclear whether the computed update is still a descent direction on the original energy.  While not crucial, such a result would be reassuring and might give further insights into the method.

2) Each update requires the solution of a nonconvex, nonlinear least squares problem which is prohibitively expensive to solve. Note that such nonlinear least squares updates are already proposed in (Carreira-Perpinan & Wang, 2014). When using ReLU activation, the non smoothness might be an issue for standard nonlinear least squares solvers such as Levenberg-Marquadt.

Furthermore, the numerical results are unfortunately a bit discouraging. The experiments evaluate toy models on toy datasets and even there only a minor improvement with respect to epochs over SGD and Prox-BP is shown. Furthermore, the plots only consider epochs and not the running time. Due to the non-linear least squares problem, I assume that each epoch for the proposed method is way more costly. Therefore I consider the experimental evaluation too preliminary. A proper evaluation would require an implementation as an optimizer in state-of-the-art deep learning frameworks and a comparison with respect to running time to standard optimizers such as SGD with momentum or Adam on the GPU.

The reported performances for MNIST are surprisingly poor. Note that vanilla SGD with momentum reaches ~98.6% test set performance on such an architecture, while the overall highest reported accuracy in this paper is 98.0%. This might be due to momentum, and it would be interesting whether the proposed method could be combined with momentum or other optimizers such as Adam as in (Frerix et al. 2018).

Overall, I don't see this a practical algorithm for training deep networks and there are few theoretical results. Therefore, I cannot recommend acceptance at this stage.

To improve the paper, I would like to see an implementation of the method on the GPU in a recent deep learning framework and an evaluation on larger models / datasets. But I am doubtful this will reach competitive performance to standard optimizers. Also, it would be interesting to see how the precision in the inner nonlinear conjugate gradient solver effects outer convergence. It might be that the subproblem does not have to be solved with very high accuracy.

Minor comments:
* Missing citation: 'Difference Target Propagation', https://arxiv.org/abs/1412.7525, studies a similar type of algorithm.

**Experience Assessment:**

I have published one or two papers in this area.

**Review Assessment: Checking Correctness Of Derivations And Theory:**

I assessed the sensibility of the derivations and theory.

**Review Assessment: Checking Correctness Of Experiments:**

I assessed the sensibility of the experiments.

**Review Assessment: Thoroughness In Paper Reading:**

I read the paper at least twice and used my best judgement in assessing the paper.

---

### Official Review · AnonReviewer2 · 2019-10-22
**Official Blind Review #2**

**Rating:** 1

**Review:**

The paper introduces a novel algorithm for computing update directions for neural network's weights.
The algorithm consists of the modified backpropagation procedure where a layer's error is computed using implicitly-updated weights.

The proposed idea is interesting, but its presentation and evaluation could be significantly improved.
First, it is not very clear what motivates the exact form of Semi-implicit BP.
Second, I find the notation a bit cumbersome, especially intermediate ^{k+1/2} updates.
I also suspect that eq. 9 contains an error, probably the l.h.s. of the first part should be \delta_i^{k+1}?
Algorithm 1 is not very helpful because only the forward pass is explained in detail and a reader must refer to the main text to understand the backward pass.

The experimental evaluation also raises a number of questions.
1) Why did authors chose only one value of the learning rate and the lambda hyperparameter? A more appropriate comparison would require slightly more extensive hyperparameter search as it may well be that ProxBP would work better with different values.
2) It is also unclear if 5 CG iterations is enough to solve the intermediate problem. Also all convergence guarantees are only provided for the exact implicit update, so at least one should ensure it is computed well enough.
3) Isn't it suspucious that ProxBP performed so bad compared to other methods on MNIST?
4) Should not the set of baselines include more advanced optimizers such as RMSProp, Adam etc? They don't seem to add more computational burden than Semi-implicit BP.

I also think it is worth discussing/investigating if the obtained update directions can be used in other gradient-based optimizers instead of pure gradients and if it can have any advantages.

Overall, it does not feel to me that the paper is ready for publication.


**Experience Assessment:**

I have published one or two papers in this area.

**Review Assessment: Checking Correctness Of Derivations And Theory:**

I assessed the sensibility of the derivations and theory.

**Review Assessment: Checking Correctness Of Experiments:**

I assessed the sensibility of the experiments.

**Review Assessment: Thoroughness In Paper Reading:**

I read the paper at least twice and used my best judgement in assessing the paper.

---

### Official Review · AnonReviewer3 · 2019-10-22
**Official Blind Review #3**

**Rating:** 3

**Review:**

This paper proposes an implicit update scheme for the back propagation a anlgorithm.
The idea is quite simple and is based on proximal mappings that lead to implicit update.
Specifically, every update in the back propagation algorithm is being replaced by an implicit update except for the intermediate parameters that receive a "semi-implicit" update.

The idea is reasonable and seems to lead to good performance. This is more-or-less expected thanks to the superior performance of implicit updates in general. So, it's good that the authors could make this work in the context of deep nets as well. Here are some more critical thoughts about the paper:


1) There is not much theoretical justification about the idea in the paper. Proposition 1 is a simple argument about the fixed point of the procedure. The argument could be made more rigorous, right now it is a bit of a sketch.
Apart from Proposition 1, there is no more theory offered. The authors could appeal in the theory of implicit SGD for that, e.g., [1,2,3,4]. This theory suggests a lot of stability properties for the implicit SGD update of Equation (27).

2) Somewhat related to (1), the authors could make a more clear connection to prior work.
For example, there is not mention of a very similar idea of "implicit back propagation" [5].
Also the literature in implicit SGD procedures is highly relevant.

3) Can we explain the results in Table 1 theoretically?



[1] Bertsekas, "Incremental proximal methods for large scale convex optimization. Mathematical
programming", 2011
[2] Kulis and Bartlett, "Implicit online learning", 2010
[3] Toulis and Airoldi, "Asymptotic and finite-sample properties of estimators
based on stochastic gradients", 2017
[4] Toulis, Airoldi, Rennie, "Statistical analysis of stochastic gradient methods for generalized linear models", 2014
[5] Fagan and Iyengar, "Robust Implicit Backpropagation", 2018

**Experience Assessment:**

I have published one or two papers in this area.

**Review Assessment: Checking Correctness Of Derivations And Theory:**

I carefully checked the derivations and theory.

**Review Assessment: Checking Correctness Of Experiments:**

I assessed the sensibility of the experiments.

**Review Assessment: Thoroughness In Paper Reading:**

I read the paper thoroughly.

---

### Decision · Program_Chairs · 2019-12-19

**Decision:**

Reject

**Comment:**

The reviewers equivocally reject the paper, which is mostly experimental and the results of which are limited.  The authors do not react to the reviewers' comments.